# COLLAPSED AMORTIZED VARIATIONAL INFERENCE FOR SWITCHING NONLINEAR DYNAMICAL SYSTEMS

## ABSTRACT

We propose an efficient inference method for switching nonlinear dynamical systems. The key idea is to learn an inference network which can be used as a proposal distribution for the continuous latent variables, while performing exact marginalization of the discrete latent variables. This allows us to use the reparameterization trick, and apply end-to-end training with stochastic gradient descent. We show that the proposed method can successfully segment time series data (including videos) into meaningful "regimes", by using the piece-wise nonlinear dynamics.

## 1 INTRODUCTION

Consider watching from above an airplane flying across country or a car driving through a field. The vehicle's motion is composed of straight, linear dynamics and curving, nonlinear dynamics. This is illustrated in fig. 1(a). In this paper, we propose a new inference algorithm for fitting switching nonlinear dynamical systems (SNLDS), which can be used to segment time series data as sequences of images, or lower dimensional signals, such as (x,y) locations into meaningful discrete temporal "modes" or "regimes". The transitions between these modes may correspond to the changes in internal goals of the agent (e.g., a mouse switching from running to resting, as in Johnson et al. (2016)) or may be caused by external factors (e.g., changes in the road curvature). Discovering such discrete modes is useful for scientific applications (c.f., Wiltschko et al. (2015); Linderman et al. (2019)) as well as for planning in the context of hierarchical reinforcement learning (c.f., Kipf et al. (2019)).

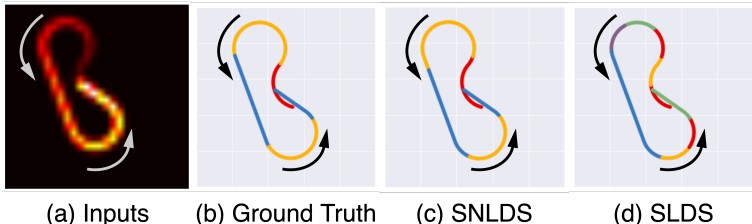

(a) Inputs    (b) Ground Truth    (c) SNLDS    (d) SLDS

Figure 1: (a): Trajectory of a particle moving along a counter-clockwise route. The direction of motion is indicated by the arrow and the brightness, from lower to brighter intensity. (b) Ground truth segmentation into "regimes". Blue is moving straight, yellow is turning counter-clockwise, red is turning clockwise. (c) Segmentation learned by our SNLDS model. (d) Segmentation learned by baseline SLDS model. Note that to model the nonlinear dynamics, the SLDS model needs to use more segments.

There has been extensive previous work, some of which we review in Section 2, on modeling temporal data using various forms of state space models (SSM). We are interested in the class of SSM which has both discrete and continuous latent variables, which we denote by $s_t$ and $z_t$, where $t$ is the discrete time index. The discrete state, $s_t \in \{1, 2, \dots, K\}$, represents the mode of the system at time $t$, and the continuous state, $z_i \in \mathbb{R}^H$, represents other factors of variation, such as location and velocity. The observed data is denoted by $x_t \in \mathbb{R}^D$, and can either be a low dimensional projection of $z_t$, such as the current location, or a high dimensional signal that is informative about $z_t$, such as an image. We may optionally have observed input or control signals $u_t \in \mathbb{R}^U$, which drive the system in addition to unobserved stochastic noise. We are interested in learning a generative model of the form

$p_{\boldsymbol{\theta}}(s_{1:T}, \boldsymbol{z}_{1:T}, \boldsymbol{x}_{1:T} | \boldsymbol{u}_{1:T})$ from partial observations, namely $(\boldsymbol{x}_{1:T}, \boldsymbol{u}_{1:T})$. This requires inferring the posterior over the latent states, $p_{\boldsymbol{\theta}}(s_{1:T}, \boldsymbol{z}_{1:T} | \boldsymbol{v}_{1:T})$, where $\boldsymbol{v}_t = (\boldsymbol{x}_t, \boldsymbol{u}_t)$ contains all the visible variables at time $t$. For training purposes, we usually assume that we have multiple such trajectories, possibly of different lengths, but we omit the sequence indices from our notations for simplicity. This problem is very challenging, because the model contains both discrete and continuous latent variables (a so-called "hybrid system"), and has nonlinear transition and observation models.

The main contribution of our paper is a new way to perform efficient approximate inference in this class of SNLDS models. The key observation is that, conditioned on knowing $\boldsymbol{z}_{1:T}$ as well as $\boldsymbol{v}_{1:T}$, we can marginalize out $s_{1:T}$ in linear time using the forward-backward algorithm. In particular, we can efficiently compute the gradient of the log marginal likelihood, $\nabla \sum_{s_{1:T}} \log p(s_{1:T} | \tilde{\boldsymbol{z}}_{1:T}, \boldsymbol{v}_{1:T})$, where $\tilde{\boldsymbol{z}}_{1:T}$ is a posterior sample that we need for model fitting. To efficiently compute posterior samples $\tilde{\boldsymbol{z}}_{1:T}$, we learn an amortized inference network $q_{\boldsymbol{\phi}}(\boldsymbol{z}_{1:T} | \boldsymbol{v}_{1:T})$ for the "collapsed" NLDS model $p(\boldsymbol{z}_{1:T}, \boldsymbol{v}_{1:T})$. The collapsing trick removes the discrete variables, and allows us to use the reparameterization trick for the continuous $\boldsymbol{z}$. These tricks let us use stochastic gradient descent (SGD) to learn $p$ and $q$ jointly, as explained in Section 3. We can then use $q$ as a proposal distribution inside a Rao-Blackwellised particle filter (Doucet et al., 2000), although in this paper, we just use a single posterior sample, as is common with Variational AutoEncoders (VAEs, Kingma & Welling (2014); Rezende et al. (2014)).

Although the above "trick" allows us efficiently perform inference and learning, we find that in challenging problems (e.g., when the dynamical model $p(\boldsymbol{z}_t | \boldsymbol{z}_{t-1}, \boldsymbol{v}_t)$ is very flexible), the model ignores the discrete latent variables, and does not perform mode switching. This is a form of "posterior collapse", similar to VAEs, where powerful decoders can cause the latent variables to be ignored, as explained in Alemi et al. (2018). Our second contribution is a new form of posterior regularization, which prevents the aforementioned problem and results in a significantly improved segmentation.

We apply our method, as well as various existing methods, to two previously proposed low-dimensional time series segmentation problems, namely a 1d bouncing ball, and a 2d moving arm. In the 1d case, the dynamics are piecewise linear, and all methods perform perfectly. In the 2d case, the dynamics are piecewise nonlinear, and we show that our method infers much better segmentation than previous approaches for comparable computational cost. We also apply our method to a simple new video dataset (see fig. 1 for an example), and find that it performs well, provided we use our proposed regularization method.

In summary, our main contributions are

- Learning switching nonlinear dynamical systems parameterized with neural networks by marginalizing out discrete variables.
- Using entropy regularization and annealing to encourage discrete state transitions.
- Demonstrating that the discrete states of nonlinear models are more interpretable.

## 2    RELATED WORK

In this section, we briefly summarize some related work.

### 2.1    STATE SPACE MODELS

We consider the following state space model:

$$p_{\theta}(\boldsymbol{x}, \boldsymbol{z}, \boldsymbol{s}) = p(\boldsymbol{x}_1 | \boldsymbol{z}_1) p(\boldsymbol{z}_1 | s_1) \left[ \prod_{t=2}^{T} p(\boldsymbol{x}_t | \boldsymbol{z}_t) p(\boldsymbol{z}_t | \boldsymbol{z}_{t-1}, s_t) p(s_t | s_{t-1}, \boldsymbol{x}_{t-1}) \right], \tag{1}$$

where $s_t \in \{1, \ldots, K\}$ is the discrete hidden state, $\boldsymbol{z}_t \in \mathbb{R}^L$ is the continuous hidden state, and $\boldsymbol{x}_t \in \mathbb{R}^D$ is the observed output, as in fig. 2(a). For notational simplicity, we ignore any observed inputs or control signals $\boldsymbol{u}_t$, but these can be trivially added to our model.

Note that the discrete state influences the latent dynamics $\boldsymbol{z}_t$, but we could trivially make it influence the observations $\boldsymbol{x}_t$ as well. More interesting are which edges we choose to add as parents of the discrete state $s_t$. We consider the case where $s_t$ depends on the previous discrete state, $s_{t-1}$, as in a

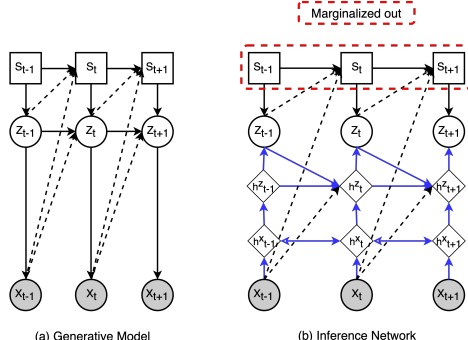

(a) Generative Model  (b) Inference Network

Figure 2: **Left:** Illustration of the generative model. Dashed arrows indicate optional connections. **Right:** Illustration of the inference network. Solid black arrows share parameters $\boldsymbol{\theta}$ with the generative model, solid blue arrows have parameters $\boldsymbol{\phi}$ that are unique to $q$. The diamonds represent deterministic nodes computed with RNNs: $h_t^x$ is a bidirectional RNN applied to $\boldsymbol{x}_{1:T}$, and $h_t^z$ is a unidirectional RNN applied to $\boldsymbol{h}_{t-1}^x$ and $\boldsymbol{z}_{t-1}$.

hidden Markov model (HMM), but also depends on the previous observation, $\boldsymbol{x}_{t-1}$. We can trivially depend on multiple previous observations; we assume first-order Markov for simplicity. This means that state changes do not have to happen "open loop", but instead may be triggered by signals from the environment. We can also condition $\boldsymbol{z}_t$ on $\boldsymbol{x}_{t-1}$, and $s_t$ on $\boldsymbol{z}_{t-1}$. It is straightforward to handle such additional dependencies (shown by dashed lines in fig. 2(a)) in our inference method, which is not true for some of the other methods we discuss below.

We still need to specify the functional forms of the conditional probability distributions. In this paper, we make the following fairly weak assumptions:

$$p(\boldsymbol{x}_t|\boldsymbol{z}_t) = \mathcal{N}(\boldsymbol{x}_t|f_x(\boldsymbol{z}_t), \mathbf{R}), \tag{2}$$

$$p(\boldsymbol{z}_t|\boldsymbol{z}_{t-1}, s_t = k) = \mathcal{N}(\boldsymbol{z}_t|f_z(\boldsymbol{z}_{t-1}, k), \mathbf{Q}), \tag{3}$$

$$p(s_t|s_{t-1} = j, \boldsymbol{x}_{t-1}) = \mathrm{Cat}(s_t|\mathcal{S}(f_s(\boldsymbol{x}_{t-1}, j))), \tag{4}$$

where $f_{x,z,s}$ are nonlinear functions (MLPs or RNNs), $\mathcal{N}(\cdot, \cdot)$ is a multivariate Gaussian distribution, $\mathrm{Cat}(\cdot)$ is a categorical distribution, and $\mathcal{S}(\cdot)$ is a softmax function. $\mathbf{R} \in \mathbb{R}^{D \times D}$ and $\mathbf{Q} \in \mathbb{R}^{H \times H}$ are learned covariance matrices for the Gaussian emission and transition noise.

If $f_x$ and $f_z$ are both linear, and $p(s_t|s_{t-1})$ is first-order Markov without dependence on $\boldsymbol{z}_{t-1}$, the model is called a switching linear dynamical system (SLDS). If we allow $s_t$ to depend on $\boldsymbol{z}_{t-1}$, the model is called a recurrent SLDS (Linderman et al., 2017; Linderman & Johnson, 2017). We will compare to rSLDS in our experiments.

If $f_z$ is linear, but $f_x$ is nonlinear, the model is sometimes called a "structured variational autoencoder" (SVAE) (Johnson et al., 2016), although that term is ambiguous, since there are many forms of structure. We will compare to SVAEs in our experiments.

If $f_z$ is a linear function, the model may need to use lots of discrete states in order to approximate the nonlinear dynamics, as illustrated in fig. 1(d). We therefore allow $f_z$ (and $f_x$) to be nonlinear. The resulting model is called a switching nonlinear dynamical system (SNLDS), or Nonlinear Regime-Switching State-Space Model (RSSSM) (Chow & Zhang, 2013). Prior work typically assumes $f_z$ is a simple nonlinear model, such as polynomial regression. If we let $f_z$ be a very flexible neural network, there is a risk that the model will not need to use the discrete states at all. We discuss a solution to this in Section 3.3.

The discrete dynamics can be modeled as a semi-Markov process, where states have explicit durations (see e.g., Duong et al. (2005); Chiappa (2014)). One recurrent, variational version is the recurrent hidden semi-Markov model (rHSMM, Dai et al. (2017)). Rather than having a stochastic continuous variable at every timestep, rHSMM instead stochastically switches between states with deterministic dynamics. The semi-Markovian structures in this work have an explicit maximum duration, which makes them less flexible. A revised method, (Kipf et al., 2019), is able to better handle unknown

durations, but produces a potentially infinite number of distinct states, each with deterministic dynamics. The deterministic dynamics of these works may limit their ability to handle noise.

## 2.2 Variational inference and learning

A common approach to learning latent variable models is to maximize the evidence lower bound (ELBO) on the log marginal likelihood (see e.g., Blei et al. (2016)). This is given by $\log p(\boldsymbol{x}) \leq \mathcal{L}(\boldsymbol{x}; \boldsymbol{\theta}, \boldsymbol{\phi}) = \mathbb{E}_{q_{\boldsymbol{\phi}}(\boldsymbol{z}, \boldsymbol{s}|\boldsymbol{x})}\left[\log p_{\boldsymbol{\theta}}(\boldsymbol{x}, \boldsymbol{z}, \boldsymbol{s}) - \log q_{\boldsymbol{\phi}}(\boldsymbol{z}, \boldsymbol{s}|\boldsymbol{x})\right]$, where $q_{\boldsymbol{\phi}}(\boldsymbol{z}, \boldsymbol{s}|\boldsymbol{x})$ is an approximate posterior.[1] Rather than computing $q$ using optimization for each $\boldsymbol{x}$, we can train an inference network, $f_{\boldsymbol{\phi}}(\boldsymbol{x})$, which emits the parameters of $q$. This is known as "amortized inference" (see e.g., Kingma & Welling (2014)).

If the posterior distribution $q_{\boldsymbol{\phi}}(\boldsymbol{z}, \boldsymbol{s}|\boldsymbol{x})$ is reparameterizable, then we can make the noise independent of $\boldsymbol{\phi}$, and hence apply the standard SGD to optimize $\boldsymbol{\theta}, \boldsymbol{\phi}$. Unfortunately, the discrete distribution $p(\boldsymbol{s}|\boldsymbol{x})$ is not reparameterizable. In such cases, we can either resort to higher variance methods for estimating the gradient, such as REINFORCE, or we can use continuous relaxations of the discrete variables, such as Gumbel Softmax (Jang et al., 2017), Concrete (Maddison et al., 2017b), or combining both, such as REBAR (Tucker et al., 2017). We will compare against a Gumbel-Softmax version of SNLDS in our experiments. The continuous relaxation approach was applied to SLDS models in (Becker-Ehmck et al., 2019) and HSSM models in (Liu et al., 2018a; Kipf et al., 2019). However, the relaxation can lose many of the benefits of having discrete variables (Le et al., 2019). Relaxing the distribution to a soft mixture of dynamics results in the Kalman VAE (KVAE) model of Fraccaro et al. (2017). A concern is that soft models may use a mixture of dynamics for distinct ground truth states rather than assigning a distinct mode of dynamics at each step as a discrete model must do. We will compare to KVAE in our experiments. In Section 3, we propose a new method to avoid these issues, in which we collapse out $\boldsymbol{s}$ so that the entire model is differentiable.

The SVAE model of Johnson et al. (2016) also uses the forward-backward algorithm to compute $q(\boldsymbol{s}|\boldsymbol{v})$; however, they assume the dynamics of $\boldsymbol{z}$ are linear Gaussian, so they can apply the Kalman smoother to compute $q(\boldsymbol{z}|\boldsymbol{v})$. Assuming linear dynamics can result in over-segmentation, as we have discussed. A forward-backward algorithm is applied once to the discrete states and once to the continuous states to compute a structured mean field posterior $q(\boldsymbol{z})q(\boldsymbol{s})$. In contrast, we perform approximate inference for $\boldsymbol{z}$ using one forward-backward pass and then exact inference for $\boldsymbol{s}$ using a second pass, as we explain in Section 3.

## 2.3 Monte Carlo inference

There is a large literature on using sequential Monte Carlo methods for inference in state space models (see e.g., Doucet & Johansen (2011)). When the model is nonlinear (as in our case), we may need a lot of particles to get a good approximation, which can be expensive. We can often get better (lower variance) approximations by analytically marginalizing out some of the latent variables; the resulting method is called a "Rao Blackwellised particle filter" (RBPF).

Prior work (e.g., Doucet et al. (2001)) has applied RBPF to SLDS models, leveraging the fact that it is possible to marginalize out $p(\boldsymbol{z}|\boldsymbol{s}, \boldsymbol{v})$ using the Kalman filter. It is also possible to compute the optimal proposal distribution for sampling from $p(\boldsymbol{s}_t|\boldsymbol{s}_{t-1}, \boldsymbol{v})$ in this case. However, this relies on the model being conditionally linear Gaussian. By contrast, we marginalize out $p(\boldsymbol{s}|\boldsymbol{z}, \boldsymbol{v})$, so we can handle nonlinear models. In this case, it is hard to compute the optimal proposal distribution for sampling from $p(\boldsymbol{z}_t|\boldsymbol{z}_{t-1}, \boldsymbol{v})$, so instead we use variational inference to learn to approximate this.

## 3 Method

### 3.1 Inference

We use the following variational posterior: $q_{\boldsymbol{\phi}, \boldsymbol{\theta}}(\boldsymbol{z}, \boldsymbol{s}|\boldsymbol{x}) = q_{\boldsymbol{\phi}}(\boldsymbol{z}|\boldsymbol{x})p_{\boldsymbol{\theta}}(\boldsymbol{s}|\boldsymbol{z}, \boldsymbol{x})$, where $p_{\boldsymbol{\theta}}(\boldsymbol{s}|\boldsymbol{z}, \boldsymbol{x})$ is the exact posterior (under the generative model) computed using the forward-backward algorithm,

---

[1] In the case of sequential models, we can create tighter lower bounds using methods such as FIVO (Maddison et al., 2017a), although this is orthogonal to our work.

and $q_\phi(z|x)$ is defined below. To compute $q_\phi(z|x)$, we first process $x_{1:T}$ through a bidirectional RNN, whose state at time $t$ is denoted by $h_t^x$. (As noted in Krishnan et al. (2017), due to the Markov assumptions of our model, we only need a backward RNN to summarize $x_{t:T}$, rather a bidirectional RNN to summarize $x_{1:T}$, but we use the latter for simplicity.) We then use a forward (causal) RNN, whose state denoted by $h_t^z$, to compute the parameters of $q(z_t|z_{1:t-1}, x_{1:T})$, where the hidden state is computed based on $h_{t-1}^z$ and $h_t^x$. This gives the following approximate posterior: $q_\phi(z_{1:T}|x_{1:T}) = \prod_t q(z_t|z_{1:t-1}, x_{1:T}) = \prod_t q(z_t|h_t^z)$. See fig. 2(b) for an illustration.

We can draw a sample $z_{1:T} \sim q_\phi(z|x)$ sequentially, and then treat this as "soft evidence" for the HMM model. The (sample dependent) parameters used in the forward-backward algorithm are given by $A_t(j,k) = p(s_t = j|s_{t-1} = k, x_{t-1})$, $B_t(k) = p(x_t|z_t)p(z_t|z_{t-1}, s_t = k)$ for $t > 1$, $B_1(k) = p(x_1|z_1)p(z_1|s_1 = k)$, and $\pi(k) = p(s_1 = k)$. We can then compute the following posterior marginals: $\gamma_t^2(j,k) = p(s_t = k, s_{t-1} = j|x_{1:T}, z_{1:T})$ and $\gamma_t^1(k) = p(s_t = k|x_{1:T}, z_{1:T})$, which can be used to compute the gradients of the log likelihood, as we discuss below.

## 3.2 LEARNING

The evidence lower bound (ELBO) for a single sequence $x$ is given by

$$\mathcal{L}_{\text{ELBO}}(\theta, \phi) = \mathbb{E}_{q_\phi(z|x)p_\theta(s|x,z)}[\log p_\theta(x,z)p_\theta(s|x,z) - \log q_\phi(z|x)p_\theta(s|x,z)] \quad (5)$$

$$= \mathbb{E}_{q_\phi(z|x)}[\log p_\theta(x,z) - \log q_\phi(z|x)] \quad (6)$$

Because $q_\phi(z)$ is reparameterizable, we can approximate the gradient as follows:

$$\nabla_{\theta,\phi}\mathcal{L}(\theta,\phi) \approx \nabla_{\theta,\phi}\log p_\theta(x, \tilde{z}) - \nabla_\phi \log q_\phi(\tilde{z}|x) \quad (7)$$

where $\tilde{z}$ is a sample from the variational proposal $\tilde{z} \sim q_\phi(\tilde{z}_1|x_{1:T})\prod_{t=2}^T q_\phi(\tilde{z}_t|\tilde{z}_{t-1}, x_{1:T})$. The second term can be computed by applying backpropagation through time to the inference RNN. In the appendix, we show that the first term is given by

$$\sum_{t=2}^T \sum_{j,k} \gamma_t^2(j,k)\nabla[\log B_t(k)A_t(j,k)] + \sum_k \gamma_1^1(k)\nabla[\log B_1(k)\pi(k)] \quad (8)$$

## 3.3 ENTROPY REGULARIZATION AND TEMPERATURE ANNEALING

When using expressive nonlinear functions (e.g. an RNN or MLP) to model $p(z_t|z_{t-1}, s_t)$, we found that the model only used a single discrete state, analogous to posterior collpase in VAEs (see e.g., Alemi et al. (2018)). To encourage the model to utilize multiple states, we add an additional regularizing term to the ELBO that penalizes the KL divergence between the state posterior at each time step and a uniform prior $p_{\text{prior}}(s_t = k) = 1/K$ (Burke et al., 2019). We call this a cross-entropy regularizer:

$$\mathcal{L}_{\text{CE}} = \sum_{t=1}^T D_{\text{KL}}(p_{\text{prior}}(s_t)||p(s_t|z_{1:T}, x_{1:T})). \quad (9)$$

Our overall objective now becomes

$$\mathcal{L}(\theta, \phi) = \mathcal{L}_{\text{ELBO}}(\theta, \phi) - \beta\mathcal{L}_{\text{CE}}(\theta, \phi). \quad (10)$$

Note that $0 \le \mathcal{L}_{\text{CE}} \le \infty$, so we need to choose the scale of $\beta > 0$ appropriately. To further smooth the optimization problem, we apply temperature annealing to the discrete state transitions, as follows: $p(s_t = k|s_{t-1} = j, x_{t-1}) = \mathcal{S}(\frac{p(s_t=k|s_{t-1}=j,x_{t-1})}{\tau})$, where $\tau$ is the temperature.

At the beginning stage of training, $\beta$, $\tau$ are set to large values. Doing so ensures that all states are visited, and can explain the data equally well. Over time, we reduce the regularizers to 0 and temperature to 1, according to a fixed annealing schedule; this allows clusters to start to separate (c.f., Rose (1998)), as each regime learns its own local dynamical model. The overall approach is similar to multi-step pretraining, as used in prior papers such as the rSLDS paper (Linderman et al., 2017), but our approach works in a continuous end-to-end fashion.

## 4 EXPERIMENTS

In this section, we compare our method to various other methods that have been recently proposed for time series segmentation using latent variable models. Since it is hard to evaluate unsupervised learning methods, such as segmentation, we use three synthetic datasets, where we know the ground truth.

In each case, we fit the model to the data, and then estimate the most likely hidden discrete state at each time step, $\hat{s}_t = \arg\max q(s_t|\boldsymbol{x}_{1:T})$. Since the model is unidentifiable, the state labels have no meaning, so we post-process them by applying the best permutation over labels so as to maximize the $F_1$ score across frames. Here the $F_1$ score is the harmonic mean of precision and recall, $2 \times \texttt{precision} \times \texttt{recall}/(\texttt{precision} + \texttt{recall})$, where $\texttt{precision}$ is the percentage of the predictions that match the ground truth states, and $\texttt{recall}$ is the percentage of the ground truth states that match the predictions. We also compute the switching-point $F_1$ by only considering the frames where the ground truth state changes. This measure compliments the frame-wise $F_1$, because it measures the accuracy in time. Since matching the exact time of the switch point is very hard in the unsupervised setting with noisy observations, we also consider a detected change point as correct, if it occurs within some small temporal interval around the ground truth as noted in Section 4.3.

### 4.1 1D BOUNCING BALL

In this section, we use a simple dataset from Johnson et al. (2016). The data encodes the location of a ball bouncing between two walls in a one dimensional space. The initial position and velocity are random, but the wall locations are constant.

We apply our SNLDS model to this data, where $f_x$ and $f_z$ are both MLPs. We found that regularization was not necessary in this experiment. We also consider the case where $f_x$ and $f_z$ are linear (which corresponds to an SLDS model), the rSLDS model of Linderman et al. (2017), the SVAE model of Johnson et al. (2016), the Kalman VAE (KVAE) model of Fraccaro et al. (2017) and a Gumbel-Softmax version of SNLDS as described in Appendix A.2. We use the implementations of rSLDS, SVAE, and KVAE provided by the authors.

As the data is generated by a simple piece-wise linear dynamics and all models we tested learn a perfect segmentation, as shown in Figure 3(a) and Table 1. This serves as a "sanity check" that we are able to use and implement the rSLDS, SVAE, KVAE and Gumbel-Softmax SNLDS code correctly. (See also Appendix A.3 for further analysis.)

Note that the "true" number of discrete states is just 2, encoding whether the ball is moving up or down. We find that our method can learn to ignore irrelevant discrete states if they are not needed. This is presumably because we are maximizing the marginal likelihood since we sum over all hidden states, and this is known to encourage model simplicity due to the "Bayesian Occam's razor" effect (Murray & Ghahramani, 2005). By contrast, with the other methods, we had to be more careful in setting $K$.

### 4.2 2D REACHER TASK

In this section, we consider a dataset proposed in the CompILE paper (Kipf et al., 2019). The observations are sequences of 36 dimensional vectors, derived from the 2d locations of various static objects, and the 2d joint locations of a moving arm (see Appendix A.4 for details and a visualization). The ground truth discrete state for this task is the identity of the target that the arm is currently reaching for (i.e., its "goal").

We fit the same 6 models as above to this dataset. Since it is a much harder problem, we found that we needed to add regularization to our model to encourage it to switch states. Figure 3(b) visualizes the resulting segmentation (after label permutation) for a single example. We see that our SNLDS model matches the ground truth more closely than our SLDS model, as well as the rSLDS, SVAE, KVAE, and Gumbel-Softmax baselines.

To compare performance quantitatively, we evaluate the models from 5 different training runs on the same held-out dataset of size 32, and compute the $F_1$ scores. We also report the $F_1$ number from CompILE. The CompILE paper uses an iterative segmentation scheme that can detect state changes,

but it does infer what the current latent state is, so we cannot include it in Figure 3(b). As in Table 1, we find that our SNLDS method is significantly better than the other approaches.

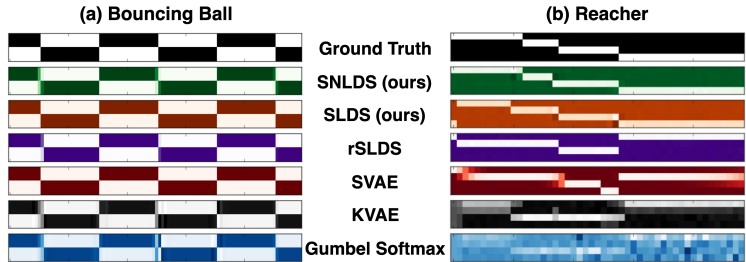

Figure 3: Segmentation on bouncing ball (left) and reacher task (right). From top to bottom: **Row** 1. ground truth of latent discrete states; **Rows** $2, 3, 4, 5, 6, 7$. the posterior marginals, $p(s_t = k | \boldsymbol{x}_{1:T}, \boldsymbol{z}_{1:T})$, of SNLDS, SLDS, rSLDS, SVAE, KVAE, and Gumbel-Softmax SNLDS respectively, where lighter color represents higher probability. CompILE is not included because it represents a different model family that directly predicts the segment boundary without calculating posterior marginals at each time step.

Table 1: Quantitative comparisons (in % $\pm \sigma$) for segmentation on bouncing ball and reacher task. We report the $F_1$ scores in percentage with mean and standard deviation over 5 runs. (*S.P.* for switching point, *F.W.* for frame-wise, the best mean is in bold.) The $F_1$ score for CompILE is adapted from Kipf et al. (2019), where only switching point $F_1$ score is provided. The $F_1$ score for KVAE is computed based on taking 'argmax' on the 'dynamics parameter network' as described in Fraccaro et al. (2017).

| DATASET | Bouncing Ball | | Reacher Task | |
|---|---|---|---|---|
| METRIC | $F_1$ (S.P.) | $F_1$ (F.W.) | $F_1$ (S.P.) | $F_1$ (F.W.) |
| **SLDS (Ours)** | 100. | 100. | $59.6 \pm 3.2$ | $81.0 \pm 3.4$ |
| **SNLDS (Ours)** | 100. | 100. | $\mathbf{78.1 \pm 4.2}$ | $\mathbf{89.0 \pm 2.0}$ |
| **rSLDS** | 100. | 100. | $47.2 \pm 3.2$ | $69.8 \pm 3.5$ |
| **SVAE** | 100. | 100. | $35.3 \pm 2.6$ | $62.3 \pm 4.9$ |
| **KVAE** | 100. | 100. | $21.5 \pm 8.0$ | $33.7 \pm 7.5$ |
| **Gumbel-Softmax SNLDS** | $97.6 \pm 1.8$ | $93.8 \pm 4.0$ | $5.0 \pm 8.7$ | $14.2 \pm 9.3$ |
| **CompILE** | - | - | $74.3 \pm 3.3$ | - |

### 4.3 IMAGE DATASET (DUBINS PATH)

In this section, we apply our method to a new dataset that is created by "rendering" a point moving in the 2d plane. The motion follows the Dubins model[2], which is a simple model for piece-wise nonlinear (but smooth) motion that is commonly used in the fields of robotics and control theory because it corresponds to the shortest path between two points that can be traversed by wheeled robots, airplanes, etc. In the Dubins model, the change in direction is determined by an external control signal $u_t$. We replace this with three latent discrete control states: go straight, turn left, and turn right. These correspond to fixed, but unobserved, input signals $u_t$ (see Appendix A.5 for details). After generating the motion, we create a series of images, where we render the location of the moving object as a small circle on a white background. Our goal in generating this dataset was to assess how well we can recover latent dynamics from image data in a very simple, yet somewhat realistic, setting.

The publicly released code for rSLDS and SVAE does not support high dimensional inputs like images (even though the SVAE has been applied to an image dataset in Johnson et al. (2016)), and there is no public code for CompILE. Therefore we could not compare to these methods for this experiment. Also, since we already showed in Section 4.2 that our method is much better than these

---

[2] https://en.wikipedia.org/wiki/Dubins_path

approaches, as well as Kalman VAE and Gumbel-Softmax version of SNLDS, on the simpler reacher task, we expect the same conclusion to hold on the harder task of segmenting videos.

Instead we focus on comparing SNLDS with SLDS to see the advantage of allowing each regime to be represented by a nonlinear model. The results of segmenting one sequence with these models using 5 states are shown in Figure 1. We see that the SLDS model has to approximate the left and right turns with multiple discrete states, whereas the non-linear model learns a more interpretable representation.

We again compare the models using $F_1$ scores in Table 2. Because the SLDS model used too many states, we calculated two versions of the metric. The first was a greedy metric that optimally assigned the best single state to match the ground truth. The second used an oracle to optimally merge states to match the ground truth. The SNLDS model significantly outperforms the SLDS in both scenarios.

Table 2: Quantitative comparisons (in %) for S(N)LDS on Dubins path. For SLDS, $F_1$ scores with both greedy 1-to-1 matching (*Greedy*) and optimal merging (*Merging*) are provided. The switching point $F_1$ scores are estimated with both precise matching (*Tol 0*) or allowing at most 5-step displacement (*Tol 5*).

| METRIC | SLDS (Greedy) | SLDS (Merging) | SNLDS |
|---|---|---|---|
| $F_1$ (Switching point, Tol 0) | $3.5 \pm 1.0$ | $4.4 \pm 3.1$ | $\mathbf{11.3 \pm 5.7}$ |
| $F_1$ (Switching point, Tol 5) | $33.7 \pm 2.5$ | $67.0 \pm 3.4$ | $\mathbf{82.5 \pm 1.9}$ |
| $F_1$ (Frame-wise) | $29.4 \pm 3.6$ | $61.5 \pm 8.0$ | $\mathbf{84.3 \pm 7.2}$ |

### 4.4 ANALYSIS OF THE ANNEALING SCHEDULE

Many latent variable models are trained in multiple stages, in order to avoid getting stuck in bad local optima. For example, to fit the rSLDS model, Linderman et al. (2017) firstly pretrain an AR-HMM and SLDS model, and then merge them; similarly, to fit the SVAE model, Johnson et al. (2016) first train with a single latent state and then increase $K$.

We found a similar strategy was necessary for the Reacher and Dubins tasks, but we do this in a smooth way using annealed regularizations. Early in training, we train with large temperature $\tau$ and entropy coefficient $\beta$. This encourages the model to use all states equally, so that the dynamics, inference, and emission sub-networks stabilized before beginning to learn specialized behavior. We then anneal the entropy coefficient to 0, and the temperature to 1 over time. We found it best to first decay the entropy coefficient $\beta$ and then decay the temperature $\tau$.

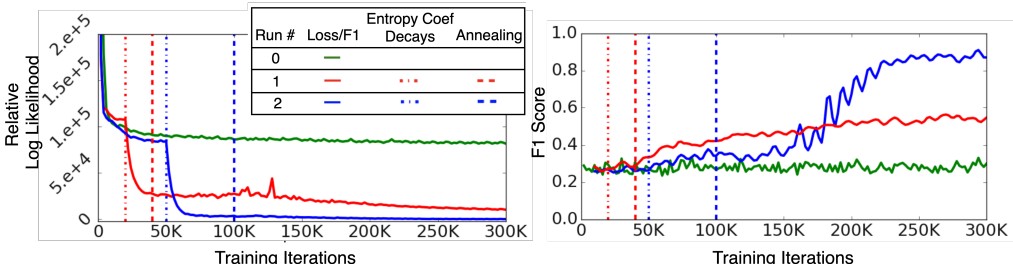

Figure 4: Comparing the relative negative log-likelihood (**left**) and the frame-wise $F_1$ scores (**right**) on Dubins paths with 3 different annealing schedules. In the first run (green), the regularization coefficient and temperature start to decay at the very beginning of training. In the second run (red), the cross entropy regularization coefficients starts to decay at step $20,000$, while temperature annealing starts at step $40,000$. In the third run (blue), the coefficient decay starts at step $50,000$, while temperature annealing starts at step $100,000$.

Figure 4 demonstrates the effect of 3 different annealing schedules on the relative log likelihood (defined as $L_t - L_{\min}$, where $L_{\min} = \min_t L_{t;1,2,3}$ across all three runs, and $L_t$ is the negative log-likelihood.), and the $F_1$ score. The green curve starts annealing right away; we see the $F_1$ score

is flat, since only one discrete state is used. The red curve starts annealing later, which improves $F_1$. However, the best results are shown in the blue curve, which starts the decay even later.

On real problems, where we have no ground truth, we cannot use the $F_1$ score as a metric to determine the best annealing schedule. However, it seems that the schedules that improve $F_1$ the most also improve likelihood the most.

## 5 CONCLUSION

We have demonstrated that our proposed method can effectively learn to segment high dimensional sequences into meaningful discrete regimes. Future work includes applying this to harder image sequences, and to hierarchical reinforcement learning.

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

## A  Appendix

### A.1  Derivation of the gradient of the ELBO

The evidence lower bound objective (ELBO) of the model is defined as:

$$
\begin{aligned}
\mathcal{L}(\theta, \phi) &= \mathbb{E}_{q_{\theta,\phi}(\boldsymbol{z}, \boldsymbol{s}|\boldsymbol{x})}\left[\log p_\theta(\boldsymbol{x}, \boldsymbol{z}, \boldsymbol{s}) - \log q_{\theta,\phi}(\boldsymbol{z}, \boldsymbol{s}|\boldsymbol{x})\right] & (11) \\
&= \mathbb{E}_{q_\phi(\boldsymbol{z}|\boldsymbol{x})p_\theta(\boldsymbol{s}|\boldsymbol{x},\boldsymbol{z})}\left[\log p_\theta(\boldsymbol{x}, \boldsymbol{z})p_\theta(\boldsymbol{s}|\boldsymbol{x}, \boldsymbol{z}) - \log q_\phi(\boldsymbol{z}|\boldsymbol{x})p_\theta(\boldsymbol{s}|\boldsymbol{x}, \boldsymbol{z})\right] & (12) \\
&= \mathbb{E}_{q_\phi(\boldsymbol{z}|\boldsymbol{x})}\left[\log p_\theta(\boldsymbol{x}, \boldsymbol{z})\right] + H(q_\phi(\boldsymbol{z}|\boldsymbol{x})) & (13)
\end{aligned}
$$

where the first term is the model likelihood, and the second is the conditional entropy for variational posterior of continuous hidden states. We can approximate the entropy of $q_\phi(\boldsymbol{z}|\boldsymbol{x})$ as:

$$
H(q_\phi(\boldsymbol{z}|\boldsymbol{x})) = H(q_\phi(\boldsymbol{z}_1)) + \sum_{t=2}^{T} H(q_\phi(\boldsymbol{z}_t|\tilde{\boldsymbol{z}}_{1:t-1})) \tag{14}
$$

where $\tilde{\boldsymbol{z}}_t \sim q(\boldsymbol{z}_t)$ is a sample from the variational posterior. In other words, we compute the marginal entropy for the output of the RNN inference network at each time step, and then sample a single latent vector to update the RNN state for the next step.

In order to apply stochastic gradient descent for end-to-end training, the minibatch gradient for the first term in the ELBO (Eq. 13) with respect to $\boldsymbol{\theta}$ is estimated as

$$
\nabla_\theta \mathbb{E}_{q_\phi(\boldsymbol{z}|\boldsymbol{x})}\left[\log p_\theta(\boldsymbol{x}, \boldsymbol{z})\right] = \mathbb{E}_{q_\phi(\boldsymbol{z}|\boldsymbol{x})}\left[\nabla_\theta \log p_\theta(\boldsymbol{x}, \boldsymbol{z})\right] \tag{15}
$$

For the gradient with respect to $\boldsymbol{\phi}$, we can use the reparameterization trick to write

$$
\nabla_\phi \mathbb{E}_{q_\phi(\boldsymbol{z}|\boldsymbol{x})}\left[\log p_\theta(\boldsymbol{x}, \boldsymbol{z})\right] = \mathbb{E}_{\boldsymbol{\epsilon} \sim N}\left[\nabla_\phi \log p_\theta(\boldsymbol{x}, \boldsymbol{z}_\phi(\boldsymbol{\epsilon}, \boldsymbol{x}))\right] \tag{16}
$$

Therefore, the gradient is expressed as:

$$
\nabla_\theta \mathcal{L} = \mathbb{E}_{q_\phi(\boldsymbol{z}|\boldsymbol{x})}\left[\nabla_\theta \log p_\theta(\boldsymbol{x}, \boldsymbol{z})\right], \tag{17}
$$

$$
\nabla_\phi \mathcal{L} = \mathbb{E}_{\boldsymbol{\epsilon} \sim N}\left[\nabla_\phi \log p_\theta(\boldsymbol{x}, \boldsymbol{z}_\phi(\boldsymbol{\epsilon}, \boldsymbol{x}))\right] + \nabla_\phi H(q_\phi(\boldsymbol{z}|\boldsymbol{x})). \tag{18}
$$

To compute the derivative of the log-joint likelihood $\nabla_{\boldsymbol{\theta},\boldsymbol{\phi}} \log p_\theta(\boldsymbol{v})$, where we define $\boldsymbol{v} = (\boldsymbol{x}_{1:T}, \boldsymbol{z}_{1:T})$ as the visible variables for brevity. Therefore

$$
\begin{aligned}
\nabla \log p(\boldsymbol{v}) &= \mathbb{E}_{p(\boldsymbol{s}|\boldsymbol{v})}\left[\nabla \log p(\boldsymbol{v})\right] & (19) \\
&= \mathbb{E}_{p(\boldsymbol{s}|\boldsymbol{v})}\left[\nabla \log p(\boldsymbol{v}, \boldsymbol{s})\right] - \mathbb{E}_{p(\boldsymbol{s}|\boldsymbol{v})}\left[\nabla \log p(\boldsymbol{s}|\boldsymbol{v})\right] & (20) \\
&= \mathbb{E}_{p(\boldsymbol{s}|\boldsymbol{v})}\left[\nabla \log p(\boldsymbol{v}, \boldsymbol{s})\right] - 0 & (21)
\end{aligned}
$$

where we used the fact that $\log p(\boldsymbol{v}) = \log p(\boldsymbol{v}, \boldsymbol{s}) - \log p(\boldsymbol{s}|\boldsymbol{v})$ and

$$
\mathbb{E}_{p(\boldsymbol{s}|\boldsymbol{v})}\left[\nabla \log p(\boldsymbol{s}|\boldsymbol{v})\right] = \int p(\boldsymbol{s}|\boldsymbol{v}) \frac{\nabla p(\boldsymbol{s}|\boldsymbol{v})}{p(\boldsymbol{s}|\boldsymbol{v})} = \nabla \int p(\boldsymbol{s}|\boldsymbol{v}) = \nabla 1 = 0. \tag{22}
$$

For $\nabla \log p(\boldsymbol{v}, \boldsymbol{s})$, we use the Markov property to rewrite it as:

$$\nabla \log p(\boldsymbol{v}, \boldsymbol{s}) = \sum_{t=2}^{T} \nabla \log p(\boldsymbol{x}_t|\boldsymbol{z}_t)p(\boldsymbol{z}_t|\boldsymbol{z}_{t-1}, s_t)p(s_t|s_{t-1}, \boldsymbol{x}_{t-1})$$
$$+ \nabla \log p(\boldsymbol{x}_1|\boldsymbol{z}_1)p(\boldsymbol{z}_1|s_1)p(s_1), \quad (23)$$

with the expectation being:

$$\nabla \log p(\boldsymbol{v}) = \mathbb{E}_{p(\boldsymbol{s}|\boldsymbol{v})}\left[\nabla \log p(\boldsymbol{v}, \boldsymbol{s})\right]$$

$$= \sum_{k} p(s_1 = k|\boldsymbol{v})\nabla \log p(\boldsymbol{x}_1|z_1)p(\boldsymbol{z}_1|s_1 = k)p(s_1 = k)$$

$$+ \sum_{t=2}^{T} \sum_{j,k} p(s_{t-1} = j, s_t = k|\boldsymbol{v})\nabla \log p(\boldsymbol{x}_t|\boldsymbol{z}_t)p(\boldsymbol{z}_t|\boldsymbol{z}_{t-1}, s_t = k)p(s_t = k|s_{t-1} = j, \boldsymbol{x}_{t-1})$$

$$= \sum_{t=2}^{T} \sum_{j,k} \gamma_t^2(j, k)\nabla \log B_t(k)A_t(j, k) + \sum_{k} \gamma_1^1(k)\nabla \log B_1(k)\pi(k). \quad (24)$$

Therefore we reach the Eq. 8.

In summary, one step of stochastic gradient ascent for the ELBO can be implemented as Algorithm 1.

---

**Algorithm 1** SVI for Training SNLDS

---

Use Bi-RNN to compute $\boldsymbol{h}_t^x$ from $\boldsymbol{x}_{1:T}$;
Recursively sample $\boldsymbol{z}_t \sim q(\boldsymbol{z}_t|\boldsymbol{z}_{t-1}, \boldsymbol{x}_{1:T})$ using forward RNN applied to $\boldsymbol{z}_{t-1}$ and $\boldsymbol{h}_t^x$;
Compute parameters $\mathbf{A}$, $\mathbf{B}$ and $\boldsymbol{\pi}$ given $\boldsymbol{x}, \boldsymbol{z}$;
Use forwards-backwards to compute $\boldsymbol{\gamma}_{1:T}^1, \boldsymbol{\gamma}_{1:T-1}^2$ from $(\mathbf{A}, \mathbf{B}, \boldsymbol{\pi})$;
Use $\boldsymbol{\gamma}$ to compute $\nabla_{\boldsymbol{\theta}, \boldsymbol{\phi}} \log p(\boldsymbol{x}, \boldsymbol{z})$;
Take gradient step.

---

## A.2 GUMBEL-SOFTMAX SNLDS

Instead of marginalizing out the discrete states with the forward-backward algorithm, one could use a continuous relaxation via reparameterization, e.g. the Gumbel-Softmax trick (Jang et al., 2017), to infer the most likely discrete states. We call this `Gumbel-Softmax SNLDS`.

We consider the same state space model as SNLDS:

$$p_\theta(\boldsymbol{x}, \boldsymbol{z}, \boldsymbol{s}) = p(\boldsymbol{x}_1|\boldsymbol{z}_1)p(\boldsymbol{z}_1|s_1)\left[\prod_{t=2}^{T} p(\boldsymbol{x}_t|\boldsymbol{z}_t)p(\boldsymbol{z}_t|\boldsymbol{z}_{t-1}, s_t)p(s_t|s_{t-1}, \boldsymbol{x}_{t-1})\right], \quad (25)$$

where $s_t \in \{1, \ldots, K\}$ is the discrete hidden state, $\boldsymbol{z}_t \in \mathbb{R}^L$ is the continuous hidden state, and $\boldsymbol{x}_t \in \mathbb{R}^D$ is the observed output, as in Figure 2(a). The inference network for the variational posterior now predicts both $\boldsymbol{s}$ and $\boldsymbol{z}$ and is defined as

$$q_{\boldsymbol{\phi}_z, \boldsymbol{\phi}_s}(\boldsymbol{z}, \boldsymbol{s}|\boldsymbol{x}) = q_{\boldsymbol{\phi}_z}(\boldsymbol{z}|\boldsymbol{x})q_{\boldsymbol{\phi}_s}(\boldsymbol{s}|\boldsymbol{x}) \quad (26)$$

where

$$q_{\boldsymbol{\phi}_z}(\boldsymbol{z}_{1:T}|\boldsymbol{x}_{1:T}) = \prod_t q(\boldsymbol{z}_t|\boldsymbol{z}_{1:t-1}, \boldsymbol{x}_{1:T}) = \prod_t q(\boldsymbol{z}_t|\boldsymbol{h}_t)\delta(\boldsymbol{h}_t|f_{RNN}(\boldsymbol{h}_{t-1}, \boldsymbol{z}_{t-1}, \boldsymbol{h}_t^b)) \quad (27)$$

$$q_{\boldsymbol{\phi}_s}(\boldsymbol{s}_{1:T}|\boldsymbol{x}_{1:T}) = \prod_t q(s_t|s_{t-1}, \boldsymbol{x}_{1:T}) = \prod_t q_{\text{Gumbel-Softmax}}(s_t|g(\boldsymbol{h}_t^b, s_{t-1}), \tau) \quad (28)$$

where $\boldsymbol{h}_t$ is the hidden state of a deterministic recurrent neural network, $f_{RNN}(\cdot)$, which works from left ($t = 0$) to right ($t = T$), summarizing past stochastic $\boldsymbol{z}_{1:t-1}$. We also feed in $\boldsymbol{h}_t^b$, which is a bidirectional RNN, which summarizes $\boldsymbol{x}_{1:T}$. The Gumbel-Softmax distribution $q_{\text{Gumbel-Softmax}}$

takes the output of a feed-forward network $g(\cdot)$ and a softmax temperature $\tau$, which is annealed according to a fixed schedule.

The evidence lower bound (ELBO) could be written as

$$\mathcal{L}_{\text{ELBO}}(\boldsymbol{\theta}, \boldsymbol{\phi}) = \mathbb{E}_{q_{\phi_z}(\boldsymbol{z}|\boldsymbol{x}) q_{\phi_s}(\boldsymbol{s}|\boldsymbol{x})} \left[ \log p_{\boldsymbol{\theta}}(\boldsymbol{x}, \boldsymbol{z}, \boldsymbol{s}) - \log q_{\phi_z}(\boldsymbol{z}|\boldsymbol{x}) q_{\phi_s}(\boldsymbol{s}|\boldsymbol{x}) \right] \tag{29}$$

One step of stochastic gradient ascent for the ELBO can be implemented as Algorithm 2.

---

**Algorithm 2** SVI for Training Gumbel-Softmax SNLDS

---

Use Bi-RNN to compute $\boldsymbol{h}_t^x$ from $\boldsymbol{x}_{1:T}$;
Recursively sample $\boldsymbol{z}_t \sim q(\boldsymbol{z}_t | \boldsymbol{z}_{t-1}, \boldsymbol{x}_{1:T})$ using forward RNN applied to $\boldsymbol{z}_{t-1}$ and $\boldsymbol{h}_t^x$;
Recursively sample $s_t \sim q_{\text{Gumbel-Softmax}}(s_t | g(\boldsymbol{h}_t^b, s_{t-1}), \tau)$ using feedforward network applied to $s_{t-1}$ and $\boldsymbol{h}_t^x$;
Compute the likelihood for eq. (29);
Take gradient step.

---

### A.3 Details on the bouncing ball experiment

The input data for bouncing ball experiment is a set of $100000$ sample trajectories, each of which is of $100$ timesteps with its initial position randomly placed between two walls separated by a distance of $10$.. The velocity of the ball for each sample trajectory is sampled from $\mathcal{U}([-0.5, 0.5])$. The exact position of ball is obscured with Gaussian noise $\mathcal{N}(0, 0.1)$. The training is performed with batch size $32$. The evaluation is carried on a fixed, held-out subset of the data with $200$ samples. For the inference network, the bi-directional and forward RNNs are both $16$ dimensional GRU. The dimensions of discrete and continuous hidden state are set to be $3$ and $4$. For SLDS, we use linear transition for continuous states. For SNLDS, we use GRU with $4$ hidden units followed by linear transformation for continuous state transition. The model is trained with fixed learning rate of $10^{-3}$, with the Adam optimizer (Kingma & Ba, 2015), and gradient clipping by norm of $5$. for $10000$ steps.

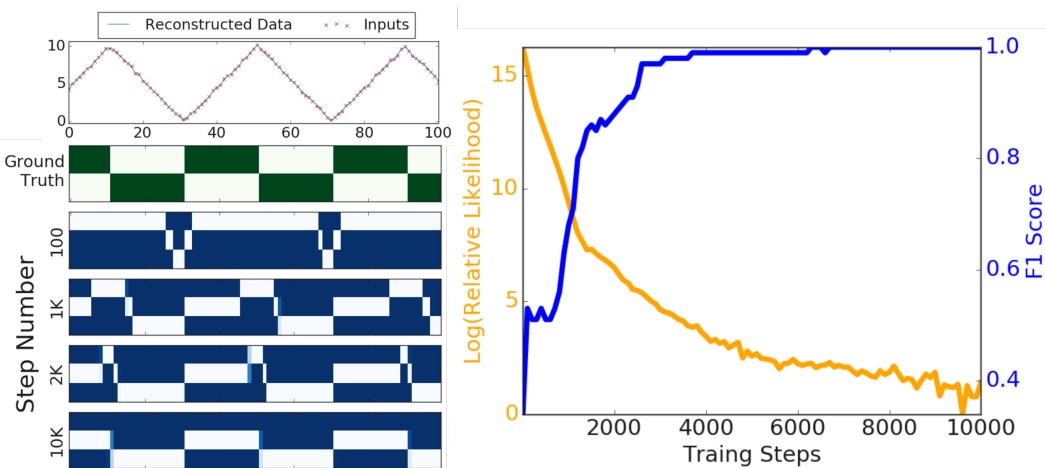

Figure 5: **Left Column:** SNLDS Segmentation on bouncing ball task with an RNN continuous transition function. Top left: illustration of input sequence and reconstruction. Center Left (green): ground truth of the latent discrete states, corresponding to two directions of motion. Lower left (blue): the posterior marginal of $p(s_t = k | \boldsymbol{x}_{1:T}, \boldsymbol{z}_{1:T})$ of SNLDS at 100, 1000, 2000 and 10000 training steps, where lighter color represents higher likelihood. **Right Column:** Training progress of relative negative log-likelihood (Orange) and frame-wise F1 score (Blue) for SNLDS. Relative negative log-likelihood is calculated as $\ln(\text{nllk} - \min(\text{nllk}) + 1.)$, where $\text{nllk}$ is negative log-likelihood. The scale emphasizes that the loss still improves even late during training.

An example of training a SNLDS model on the Bouncing Ball task is provided as Figure 5. Early in training, the discrete states do not align well to the ground truth transitions. The three states

transition rapidly near one of the walls and the frame-wise F1 score is near chance values. However, by ten thousand iterations, the model has learned to ignore one state and switches between the two states corresponding to the ball bouncing from the wall. Notably the negative log-likelihood changes by over 10 orders of magnitude before the model learns accurate segmentation of even this simple problem. We hypothesize that the likelihood is dominated by errors in continuous dynamics rather than in the discrete segmentation until very late in training.

## A.4 DETAILS ON THE REACHER EXPERIMENT

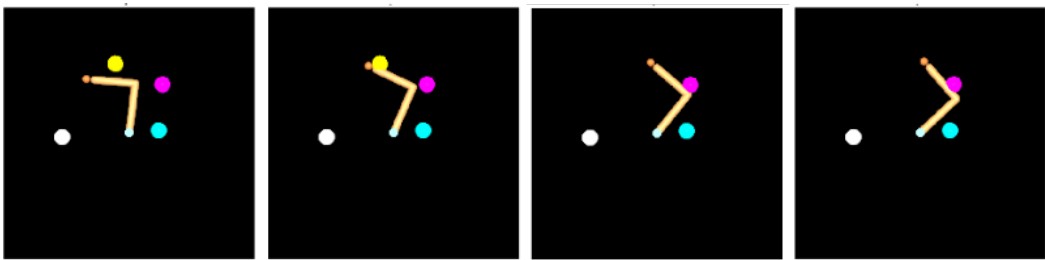

Figure 6: Illustration of the observations in reacher experiment. This is 2-D rendering of the observational vector, but the inputs to the model are sequences of vectors, as in Kipf et al. (2019), not images.

The observations in the reacher experiment are sequences of 36 dimensional vectors, as described in Kipf et al. (2019). First 30 elements are the target indicator, $\alpha$, and location, $x, y$, for 10 randomly generated objects. 3 out of 10 objects start as targets, $\alpha = 1$. The $(x, y)$ location for 5 of the non-target objects are set to $(0, 0)$. A deterministic controller moves the arm to the indicated target objects. Once a target is reached, the indicator is set to $\alpha = 0$. (Depicted as the yellow dot disappearing in Figure 6.) The remaining 6 elements of the observations are the two angles of reacher arm and the positions of two arm segment tips. The training dataset consists of 10000 observation samples, each 50 timesteps in length.

This more complex task requires more careful training. The learning rate schedule is a linear warm-up, $10^{-5}$ to $10^{-3}$ over 5000 steps, from followed by a cosine decay, with decay rate of 2000 and minimum of $10^{-5}$. Both entropy regularization coefficient starts to exponentially decay after 50000 steps, from initial value 1000 with a decay rate 0.975 and decay steps 500. The temperature annealing follows the same exponential but only starts to decay after 100000 steps. The training is performed in minibatches of size 32 for 300000 iterations using the Adam optimizer (Kingma & Ba, 2015).

The model architecture is relatively generic. The continuous hidden state $z$ is 8 dimensional. The number of discrete hidden states is set to 5 for training, which is larger than the ground truth 4 (including states targeting 3 objects and a finished state). The observations pass through an encoding network with two 256-unit ReLU activated fully-connected nets, before feeding into RNN inference networks to estimate the posterior distributions $q(z_t|x_{1:T})$. The RNN inference networks consist of a 32-unit bidirection LSTM and a 64-unit forward LSTM. The emission network is a three-layer MLP with $[256, 256, 36]$ hidden units and ReLU activation for first two layers and a linear output layer. Discrete hidden state transition network takes two inputs: the previous discrete state and the processed observations. The observations are processed by the encoding network and a 1-D convolution with 2 kernels of size 3. The transition network outputs a $5 \times 5$ matrix for transition probability $p(s_t|s_{t-1})$ at each timestep. For SNLDS, we use a single-layer MLP as the continuous hidden state transition functions $p(z_t|z_{t-1}, s_t)$, with 64 hidden units and ReLU activation. For SLDS, we use linear transitions for the continuous state.

## A.5 DETAILS ON THE DUBINS PATH EXPERIMENT

The Dubins path model[3] is a simplified flight, or vehicle, trajectory that is the shortest path to reach a target position, given the initial position $(x_0, y_0)$, the direction of motion $\theta_0$, the speed constant $V$,

---

[3] https://en.wikipedia.org/wiki/Dubins_path

and the maximum curvature constraint $\dot{\theta} \leq u$. The possible motion along the path is defined by

$$\dot{x}_t = V \cos(\theta_t), \ \dot{y}_t = V \sin(\theta_t), \ \dot{\theta}_t = u.$$

The path type can be described by three different modes/regimes: 'right turn (R)' , 'left turn (L)' or 'straight (S).'

To generate a sample trajectory used in training or testing, we randomly sample the velocity from a uniform distribution $V \sim \mathcal{U}([0.1, 0.5])$ (pixel/step), angular frequency from a uniform distribution $u/2\pi \sim \mathcal{U}([0.1, 0.15])$ (/step), and initial direction $\theta_0 \sim \mathcal{U}([0, 2\pi))$. The generated trajectories always start from the center of image $(0, 0)$. The duration of each regime is sampled from a Poisson distribution with mean 25 steps, with full sequence length 100 steps. The floating-point positional information is rendered onto a $28 \times 28$ image with Gaussian blurring with $0.3$ standard deviation to minimize aliasing.

The same schedules as in the reacher experiment are used for the learning rate, temperature annealing and regularization coefficient decay.

The network architecture is similar to the reacher task except for the encoder and decoder networks. Each observation is encoded with a CoordConv (Liu et al., 2018b) network before passing into RNN inference networks, the archicture is defined in Table 3. The emission network $p(\boldsymbol{x}_t|\boldsymbol{z}_t)$ also uses a CoordConv network as described in Table 4. The continuous hidden state $z$ in this experiment is 4 dimensional. The number of discrete hidden states $s$ is set to be 5, which is larger than ground truth 3. The inference networks are a 32-unit bidirection LSTM and a 64-unit forward LSTM. The discrete hidden state transition network takes the output of observation encoding network in the same manner as the reacher task. For SNLDS, we use a two-layer MLP as continuous hidden state transition function $p(z_t|z_{t-1}, s_t)$, with $[32, 32]$ hidden units and ReLU activation. For SLDS, we use linear transition for continuous states.

Table 3: CoordConv encoder Architecture. Before passing into the following network, the image is padded from $[28, 28, 1]$ to $[28, 28, 3]$ with the pixel coordinates.

| Layer | Filters | Shape | Activation | Stride | Padding |
|---|---|---|---|---|---|
| 1 | 2 | [5, 5] | relu | 1 | same |
| 2 | 4 | [5, 5] | relu | 2 | same |
| 3 | 4 | [5, 5] | relu | 1 | same |
| 4 | 8 | [5, 5] | relu | 2 | same |
| 5 | 8 | [7, 7] | relu | 1 | valid |
| 6 | 8 | 2 (Kernel Size) | None | 1 | causal |

Table 4: CoordConv decoder Architecture. Before passing into the following network, the input $z_t$ is tiled from $[8]$ to $[28, 28, 8]$, where 8 is the hidden dimension, and is then padded to $[28, 28, 10]$ with the pixel coordinates.

| Layer | Filters | Shape | Activation | Stride | Padding |
|---|---|---|---|---|---|
| 1 | 14 | [1, 1] | relu | 1 | valid |
| 2 | 14 | [1, 1] | relu | 1 | valid |
| 3 | 28 | [1, 1] | relu | 1 | valid |
| 4 | 28 | [1, 1] | relu | 1 | valid |
| 5 | 1 | [1, 1] | relu | 1 | same |

See Figure 7 for an illustration of the reconstruction abilities (of the observed images) for the SLDS and SNLDS models. They are visually very similar; however, the SNLDS has a more interpretable latent state as described in Section 4.3.

## A.6 REGULARIZATION AND MULTI-STEPS TRAINING

Training our SNLDS model with a powerful transition network but without regularization will fit the dynamics $p(\boldsymbol{z}_t|\boldsymbol{z}_{t-1}, s_t)$ with a single state. With randomly initialized networks, one state fits the

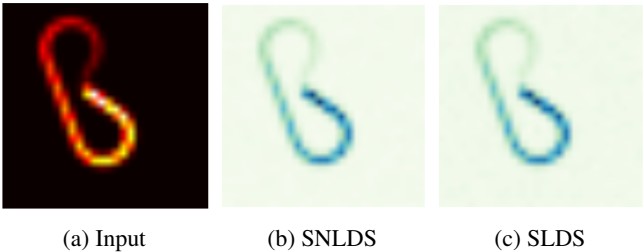

(a) Input        (b) SNLDS        (c) SLDS

Figure 7: Image sequence reconstruction for Dubins path. The sequence is averaged with early timepoints scaled to low intensity, late timepoints unchanged to indicate direction.

dynamics better at the beginning and the forward-backward algorithm will cause more gradients to flow through that state than others. The best state is the only one that gets better.

To prevent this, we use regularization to cause the model to select each mode equally likely until the inference and emission network are well trained. Thus all discrete modes are able to learn the dynamics equally well initially. When the regularization decays, the transition dynamics of each mode can then specialize. The regularization helps the model to better utilize its capacity, and the model can achieve better likelihood, as demonstrated in Section 4.4 and Figure 4.

Multi-steps training has been used by previous models, and it serves the same purpose as our regularization. SVAE first trains a single transition model, then uses that one set of parameters to initialize all the transition dynamics for multiple states in next stage of training. rSLDS training begins by fitting a single AR-HMM for initialization, then fits a standard SLDS, before finally fitting the rSLDS model. We follow these implementations of both SVAE and rSLDS in our paper. Both multi-step training and our regularization ensure the hidden dynamics are well learned before learning the segmentation. What makes regularization interesting is that it allows the model to be trained with a smooth transition between early and late training.

