# OpenReview forum: "Collapsed amortized variational inference for switching nonlinear dynamical systems"
_ICLR.cc/2020/Conference — Reject_

### Official Review · AnonReviewer1 · 2019-10-21
**Official Blind Review #1**

**Rating:** 3

**Review:**

Thank you for an interesting read.

As far as I understand, the paper claims two contributions:
1. A combination of collapsed variational inference and amortised inference for SNLDS, to make the training pipeline fully differentiable;
2. An improved loss function upon the variational lowerbound (ELBO) to force the model to use the discrete states.

======= novelty =======
The 1st idea is combinatorial:
1. The forward-backward algorithm is a standard inference method for HMM-like sequence models; collapsed variational inference has been investigated extensively in 2000s when hierarchical Bayes models were actively developed; amortised inference is widely used in variational auto-encoders.
2. The combination of the above two inference methods on S(N)LDS is new to the best of my knowledge. However, this combination has been proposed on a similar model called Kalman VAE ( Fraccaro et al. 2017), where the sequence model can be viewed as a "soft" version of SLDS.

The 2nd idea is interesting but not very well explained to some extent:
1. The goal of the modified objective function is to encourage the model to use the discrete states (instead of pushing all useful information to the continuous states). It is interesting as it regularises the *exact posterior* of the discrete states conditioned on the *approximately inferred* continuous states.
2. I believe the entropy regulariser is non-differentiable as it is based on a *histogram* estimate of the temporally averaged discrete state distribution. How exactly is this regulariser implemented?
3. I agree adding the KL regulariser can avoid the iterating assignment pathology, however, is random assignment of the regime preferred in any case? From the introductory example, I think contiguous segments are preferred.

======= significance =======
Experiments consider 3 synthetic examples for sequence segmentation (so that ground truth is available). The proposed approach performs significantly better which is a good sign. The paper also provides useful analysis on the effects of balancing parameter tempering which is always welcome.

However, two baselines are missing:
1. To claim the significance of the collapsed variational inference approach, a non-collapsed inference version of the proposed SNLDS model needs to be compared. The authors did discuss this and mentioned possible workarounds (e.g. using the Gumbel-softmax trick for discrete state inference), but the comparison is not reported. If compared, this will serve well as an ablation study for the inference method.
2. The paper also provides comparisons across models, but I do think the Kalman VAE model needs to be compared to SNLDS. Both models are more flexible than the original SLDS, but the complexity is added in different ways. Since I think the inference mechanisms are similar (both using forward-backward inference for top-level latents and amortised inference for bottom-level latents), this comparison would provide a better ablation study on the modelling side.

======= clarity =======
1. The paper presentation is overall clear to me, although I found the many sentences in parenthesis a bit distracted, so I would suggest maybe using footnotes for them instead.
2. For readers who are less familiar with HMMs/forward-backward algorithms, the papers can be difficult to understand, as it skips all the detailed computation of the gamma terms. I would suggest adding the details in the appendix, and/or visualise the intuition using e.g. message passing on factor graphs.
3. I found the related work well presented with most relevant papers, although I do think the Kalman VAE approach is highly relevant which needs to be cited and discussed.

======== references ========
Fraccaro et al. (2017). A Disentangled Recognition and Nonlinear Dynamics Model for Unsupervised Learning. NeurIPS 2017

**Experience Assessment:**

I have published one or two papers in this area.

**Review Assessment: Checking Correctness Of Derivations And Theory:**

I carefully checked the derivations and theory.

**Review Assessment: Checking Correctness Of Experiments:**

I assessed the sensibility of the experiments.

**Review Assessment: Thoroughness In Paper Reading:**

I read the paper at least twice and used my best judgement in assessing the paper.

---

> ### Author Response · Authors · 2019-11-15
> **Reply to Reviewer #1**
>
> Thank you for comments! We have revised the paper to address your comments and provide further clarifications for each point below:
>
>
> --“this combination has been proposed on a similar model called Kalman VAE ( Fraccaro et al. 2017), where the sequence model can be viewed as a "soft" version of SLDS.”; “although I do think the Kalman VAE approach is highly relevant which needs to be cited and discussed.”; and “I do think the Kalman VAE model needs to be compared to SNLDS.”:
>
>       Excellent suggestion! We have added discussion and evaluation of KVAE in the paper (see the revised Table 1 & Figure 3). KVAE segmented the bouncing ball task well (F1=1.0) but segmented the reacher task poorly (F1=0.21) with or without temperature and regularization annealing (and using hyperparameter tuning). The KVAE model uses a soft mixture of linear dynamics to represent transitions, rather than our hard nonlinear dynamics.
>
> --“a non-collapsed inference version of the proposed SNLDS model needs to be compared.”:
>
>       Excellent suggestion! We have added a discussion and evaluation of Gumbel-softmax to the paper (see the revised Table 1 & Figure 3). It  learns to segment the bouncing ball task well (F1=0.98) but segmented the reacher task poorly (F1=0.05) even with temperature and regularization annealing - its posterior segmentation is a "blurry mess" (see new figure 3), consistent with other work (eg Le'19)
>
> Reference:
>     T. A. Le, A. R. Kosiorek, N. Siddharth, Y. W. Teh, and F. Wood, “Revisiting Reweighted Wake-Sleep for Models with Stochastic Control Flow,” in UAI, 2019
>
> --“I believe the entropy regulariser is non-differentiable as it is based on a *histogram* estimate of the temporally averaged discrete state distribution. How exactly is this regulariser implemented?“:
>
>       This is a great question that prompted us to look at our implementation more carefully. Surprisingly, our implementation was silently dropping the non-differentiable terms without producing an error message. Since our implementation never used H(O), we have revised the paper to no longer mention H(0). We have verified that the results of our model are unaffected by training them again.
>
> --“I agree adding the KL regulariser can avoid the iterating assignment pathology, however, is random assignment of the regime preferred in any case? From the introductory example, I think contiguous segments are preferred.”:
>
>      You are correct that contiguous segments are preferred. The model only learns contiguous states when the regulariser becomes small due to annealing. This relationship is demonstrated in Figure 4 where the F1 only increases above chance levels after the regularizer is annealed to zero and performance only reaches peak levels when the temperature coefficient is also annealed. We have tried to make this more clear in the text and appendix (See A.6).

---

### Official Review · AnonReviewer3 · 2019-10-23
**Official Blind Review #3**

**Rating:** 8

**Review:**

In this paper, the authors consider the problem of learning model parameters of a switching nonlinear dynamical system from a dataset. They propose a new variational inference algorithm for this model-learning problem that marginalizes all discrete random variables in the model using the forward-backward algorithm and, in so doing, converts the model to one with a differentiable density, so that the gradient of the variational objective can be estimated with the low-variance reparameterization estimator. The authors also point out an issue in choosing a variational objective; the standard ELBO objective is not suitable for their learning problem, because it leads to a model that does not use discrete random variables meaningfully. To overcome this issue, they suggest a new improved objective and a learning procedure, which encourage the learned model to use discrete variables for capturing different modes of dynamics. The proposed variational inference algorithm was applied to three datasets, and in all these cases, it showed promising results.

I found the main idea and technique of the paper simple and nice. I am reasonably positive about the paper. The main text of the paper is written well, but the experimental result section seems to be rushed and needs to be polished slightly. I gave weak accept, but if the authors give a convincing answer for my question below, I may raise my score.

I presume that the objective L(theta,phi) in (11) is optimized by a version of gradient ascent. Here is my question related to this:

[Q] Why is H(O) in p5 differentiable with respect to theta and phi?

I am asking this question because the distribution O is defined in terms of arg max, which is not a differentiable operator. Furthermore, the definition of O uses p(s_t|z,x), which uses the model parameters theta. Oh, by the way, I think that the definitions of H and L_CE should include the expectation with respect to q_theta(z|x).

Some minor comments are added below.

* formula (1), p2: p(x1|s1) should be replaced by p(x1|z1)p(z1|s1)

* p3: There are no sub-figures labeled with (a) and (b) in Figure 2. I suggest to put (a) and (b) in front of the captions of the two diagrams in Figure 2. A similar comment applies to Figure 3, because the main text refers to something called Figure 3(a) and Figure 3(b). Also, the paper uses fig. 2(b) sometimes, and Figure 2(b) in other times. Using one convention consistently might help some readers.

* p3: Cat(s_t | S(f_s(...)) ===> Cat(s_t | S(f_s(...)))

* p3: SDLS ===> SLDS

* p3: log p(x) <= L(...) ===> log p(x) >= L(...)

* p4: I found the phrase "so they need to perform multiple forward-backward (FB) passes" vague. The algorithm in the paper uses FB twice, and "multiple" in the quoted phrase might mean 2, 3 or more. This makes it less clear whether the algorithm has any benefit over the existing approaches.

* p6: This measures compliments ===> This measure complements

* p6: within some small temporal around ... where noted ===> within some range around ... as noted

* p6: are is constant ===> are constant

* p6: The ground truth discrete states ===> The ground truth discrete state

* p7: The resulting of ===> The result of

**Experience Assessment:**

I have published one or two papers in this area.

**Review Assessment: Checking Correctness Of Derivations And Theory:**

I carefully checked the derivations and theory.

**Review Assessment: Checking Correctness Of Experiments:**

I assessed the sensibility of the experiments.

**Review Assessment: Thoroughness In Paper Reading:**

I read the paper thoroughly.

---

> ### Author Response · Authors · 2019-11-15
> **Reply to Reviewer #3**
>
> Thank you for the comments! We have revised the paper to address your comments and provide further clarifications for each point below:
>
> --"Why is H(O) in p5 differentiable with respect to theta and phi?“:
>
>       This is a great question that prompted us to look at our implementation more carefully. Surprisingly, our implementation was silently dropping the non-differentiable terms without producing an error message. Since our implementation never used H(O), we have revised the paper to no longer mention H(0). We have verified that the results of our model are unaffected by training them again.
>
>
> --I found the phrase "so they need to perform multiple forward-backward (FB) passes" vague. The algorithm in the paper uses FB twice, and "multiple" in the quoted phrase might mean 2, 3 or more. This makes it less clear whether the algorithm has any benefit over the existing approaches.
>
>       We have clarified this to state that SVAE has to do an FB pass on both the discrete states (to compute q(s)) and the continuous states (to compute q(z)); they say they repeat this process after re-optimizing the local evidence potentials, but indeed it is not clear how many times they do this. By contrast we do one FB pass on the biRNN to compute q(z) (which we then sample from), and one FB pass to compute q(s) given z and x. The computational cost is thus very similar. However we can handle nonlinear transition dynamics, which leads to more meaningful segmentations, as we show.

---

### Official Review · AnonReviewer2 · 2019-10-25
**Official Blind Review #2**

**Rating:** 3

**Review:**

SUMMARY:
This paper proposes a method to segment time series into discrete intervals in an unsupervised way. The data is modeled using a state space model where each state consists of a discrete and a continuous part. The discrete state denotes the segment the system is currently in and the continuous state which is conditioned on the discrete one denotes an uninterpretable feature vector. The transition distributions are non-linear. The observation at each time step is high-dimensional and produced by an emission distribution whose parameters are given by a neural network which takes in the continuous state. Learning and inference is done by maximizing the evidence lower bound (ELBO). Problems with the discreteness of latent variables is circumvented by marginalizing (collapsing) them out using the forward-backward algorithm. Problems with making discrete states meaningful when there are non-linear transitions/emissions is addressed by annealing. This annealing scheme forces the conditional distributions on the discrete state to have high entropy (be close to a uniform distribution) at the start by adding a term to the ELBO objective and the multiplier of this term is decreased as the training progresses. There are actually two terms to do this since one alone didn't work.

STRUCTURE:
The paper is well-written and easy to understand.

NOVELTY:
I found the technique of estimating gradients using forward-backward to be interesting and potentially useful in other domains when parts of generative models can be marginalized out using belief propagation.
While the problem of unsupervised time-series segmentation is an important one, I'm not sure the proposed technique addresses it completely.
The main thing that seems to be doing the work is not the marginalization using forward-backward, but rather the annealing scheme which itself seems ad-hoc and it is not clear whether this is generalizable to other domains or it just happened to work on the problems in the paper.
It is not clear why maximizing the entropy of the variational transition should encourage meaningful clustering.
Would this work even if the emission distribution is made much more powerful?

EXPERIMENTS:
There are experiments on three synthetic datasets. While the proposed method beats the competing methods, it is unclear that collapsing is helpful. Also, no annealing was used in the baseline methods (like increasing K in SVAE or multi-step training).

CONCLUSION:
While the problem this paper is tackling is significant, it isn't clear that the proposed method tackles it. I would consider bumping up my score if
- this method is demonstrated to work on a real dataset and/or
- there is a better understanding of the principles behind why this annealing scheme helps.
Also, proper tweaking of the competing algorithms (similar to annealing) is needed to compare the proposed method fairly.

**Experience Assessment:**

I have read many papers in this area.

**Review Assessment: Checking Correctness Of Derivations And Theory:**

I assessed the sensibility of the derivations and theory.

**Review Assessment: Checking Correctness Of Experiments:**

I assessed the sensibility of the experiments.

**Review Assessment: Thoroughness In Paper Reading:**

I read the paper at least twice and used my best judgement in assessing the paper.

---

> ### Author Response · Authors · 2019-11-15
> **Reply to Reviewer #2**
>
> Thank you for comments. We have revised the paper to address your comments and provide further clarifications for each point below:
>
> --“The main thing that seems to be doing the work is not the marginalization using forward-backward, but rather the annealing scheme” and “it is unclear that collapsing is helpful":
>
>       We addressed this by adding a new “ablation” result that uses the Gumbel-Softmax trick instead of marginalization (see the revised Table 1 & Figure 3). We searched the same range of hyperparameters we used to optimize the marginalized model for the reacher task. The Gumbel-Softmax model learned to segment the bouncing ball task well (F1=0.98) but segmented the reacher task poorly (F1=0.05) even with temperature and regularization annealing. As presented in the text, marginalization is critical for performance because it reduces variance of discrete optimization, and keeps sharp boundaries between segments.
>
> --“it is not clear whether this [method] is generalizable to other domains or it just happened to work on the problems in the paper.” , “ I would consider bumping up my score if demonstrated to work on a real dataset”:
>
>       We are currently working on applying our model on data sets from “real” domains. In the meantime, we used synthetic data so we can quantitatively evaluate performance,  as is common in the field. Please note that the 4 papers detailing the baseline models we compare to also mostly used synthetic data (the 2 real datasets had no ground truth segments, so evaluation was subjective).
>
> --“It is not clear why maximizing the entropy of the variational transition should encourage meaningful clustering.”, "better understanding of the principles behind why this annealing scheme helps”:
>
>       Adding a term that encourages the posterior to stay close to uniform (our L_CE term), and annealing it slowly, is a well-known method for encouraging all the discrete states to be equally "well trained" before the model starts partitioning the data into clusters. See eg Ueda'98. This can be thought of as a continuous version of multi-stage training, which is used in most prior works on SVAE and SLDS. In our paper, we also suggested a second entropy regularizer, H(O). However, in response to R3, we checked our code, and found that we were not using this as part of the objective. We have thus removed this from the paper, simplifying our method. (We checked the results are unchanged.)
>
> Reference:
>     N. Ueda and R. Nakano, “Deterministic annealing EM algorithm,” Neural Netw., vol. 11, no. 2, pp. 271–282, Mar. 1998
>
>
> --“Would this work even if the emission distribution is made much more powerful?”
>
>       Ignoring the latent state (whether discrete or continuous) is potentially a problem for any deep generative model, but this has been addressed in many other works (eg van den Oord'19).
> We could use similar methods in our case, but we did not need to, since our emission model is simple.
>
> Reference:
>     A. R. A. van den Oord Ben Poole Oriol Vinyals, “Fixing Posterior Collapse with delta-VAEs,” in ICLR, 2019
>
> --"no annealing was used in the baseline methods (like increasing K in SVAE or multi-step training).”,  “proper tweaking of the competing algorithms (similar to annealing) is needed to compare the proposed method fairly.“:
>
>       We did proper tweaking of the competing algorithms in the original submission. We have expanded the description of regularization and multi-step training in the appendix (see section A.6). We performed multi-step training for SVAE. Although not described in detail in the SVAE paper, it is a part of their reference implementation on Github. We also performed multi-step training for rSLDS. For fair comparison, we also trained the new Gumbel Softmax and KVAE models with our annealing.  Both methods segmented the bouncing ball well (F1=0.98 and 1.0 respectively), but neither model segmented the reacher task well (F1=0.05 and 0.21 respectively). See the new figure 3.

---

### Author Response · Authors · 2019-11-15
**Response to the area chair**

We thank the reviewers for their careful reading of the paper. We provide a detailed response to each reviewer  below; here, we summarize the largest changes.

First, at the prompting of R2 and R1, we have added experiments using gumbel-softmax. This is exactly the same model as ours, but uses a soft relaxation of the discrete states, instead of marginalizing them out using forwards-backwards. We also added experiments using Kalman VAE, as requested by R1; this uses a slightly different model (soft mixture of *linear* transition models), and a slightly different algorithm (it marginalizes out the continuous variables using Kalman smoother). Both methods segmented the bouncing ball well (F1=0.98 and 1.0 respectively), but neither model segmented the reacher task well (F1=0.05 and 0.21 respectively), despite our best efforts at hyperparameter tuning. We have added these results to the new version of the paper. The new figure 3 shows that the gumbel-softmax relaxation blurs all the segmentation boundaries, losing the benefits of discreteness (this is consistent with other works, such as Le'19).

Second, in response to R3 who asked how we computed gradients of the entropy regularizer H(O), through a non-continuous argmax, we looked at our (TF) implementation more carefully, and noticed it was silently dropping the non-differentiable terms without producing an error message. Since our implementation thus never used H(O), we have revised the paper to no longer mention H(0). We have verified that the results of our model are unaffected by training them again.
Third, we have clarified the presentation in a variety of ways, and added some missing references.

Reference:
    T. A. Le, A. R. Kosiorek, N. Siddharth, Y. W. Teh, and F. Wood, “Revisiting Reweighted Wake-Sleep for Models with Stochastic Control Flow,” in UAI, 2019

---

### Decision · Program_Chairs · 2019-12-19

**Decision:**

Reject

**Comment:**

This is an interesting paper on an important topic.  The reviewers identified a variety of issues both before and after the feedback period; I urge the authors to consider their comments as they continue to refine and extend their work.